# Empirically Comparing Magnetic Needle Steering Models Using Expectation-Maximization

Richard L. Pratt [†] and Andrew J. Petruska *,[†]

Mechanical Engineering Department, Colorado School of Mines, 1500 Illinois St, Golden, CO 80401, USA; rlpratt@mines.edu
* Correspondence: apetruska@mines.edu
† These authors contributed equally to this work.

**Abstract:** Straight-line needle insertion is a prevalent tool in surgical interventions in the brain, such as Deep Brain Stimulation and Convection-Enhanced Delivery, that treat a range of conditions from Alzheimer's disease to brain cancer. Using a steerable needle to execute curved trajectories and correct positional deviation could enable more intervention possibilities, while reducing the risk of complication in these procedures. This paper experimentally identifies model parameters using an expectation–maximization (EM) algorithm for two different steerable needle models. The results compared a physically motivated model to the established bicycle needle model and found the former to be preferred for modeling soft brain tissue needle insertion. The results also supported the experimentally parameterized models' use in future applications such as needle steering control.

**Keywords:** steerable needles; expectation–maximization; flexible robots; medical robots

## 1. Introduction

Needles are a longstanding and widespread tool used by surgeons as they are minimally invasive and versatile in a wide range of clinical applications. In recent years, robotically guided needles have been investigated, and one area with potential benefits is needle insertions into the brain. One such surgical intervention is Deep Brain Stimulation (DBS), which is applied to a wide range of central nervous system pathologies such as Parkinson's disease and Alzheimer's disease [1,2]. DBS surgeries place an electrode in the brain by inserting it through the skull along a straight path [3]. The required precision for an effective DBS electrode placement is estimated at 2 mm [4], and current surgical procedures achieve accuracies of about 2–3 mm [5], which means this technique can unfortunately lead to inaccurate electrode placement. This results in the need for additional electrode insertions, which reduce the likelihood of a positive surgical outcome [6]. Reducing or correcting trajectory error can reduce the number of penetrations into the brain for this procedure [7]. As most are bilateral, DBS surgeries typically require two entry points through the skull [8]. Fewer entry points would allow safer, faster, and less traumatic procedures and may be achieved by using curved trajectories to perform multiple electrode placements during a single insertion. Entry points must also be chosen to provide a straight path to the electrode placement target without penetrating sensitive areas of the brain and compromising safety. This can be difficult if not impossible depending on patient anatomy [9], whereas a curved trajectory could allow for an optimal insertion point while still allowing a safe trajectory by circumventing sensitive areas.

Another clinical application that could benefit from needle navigation around obstacles specifically in brain tissue is the treatment of cancerous brain tumors, such as glioblastoma. These tumors can develop near sensitive tissues such as venous sinuses, the brain stem, or deep cerebellar nuclei [10]. Convection-Enhanced Delivery (CED) is a targeted drug delivery technique used to treat cancerous brain tumors, as well as Parkinson's disease and

Alzheimer's disease, which, similarly to DBS, requires one or more insertions through burr holes in the skull and accurate positioning of the needle at the injection site [11].

Expanding beyond in-brain applications, there are numerous medical procedures that can benefit from steerable needle technology. A study conducted at the Annual Meeting of Cardiovascular and Interventional Radiology Society of Europe in 2016 supports the desire for steerable needles by practitioners in the field [12]. Respondents consisted primarily of interventional radiologists with experience in needle placement. The study found that added value for steerable needles in current interventions was seen by 93% of the respondents, while 85% of the respondents found needle steering to be a useful tool for steering around anatomical obstacles.

Both passive and active steering approaches to provide tip control to the surgeon have been investigated, and both present their own challenges. Passive bevel-tip needles are typically less complex, but manipulability is limited [13,14]. Active methods are typically more capable, but achieve this through sacrificing the simplicity and flexibility of the needle [15]. Magnetically steered needles may provide an ideal solution as the only added complexity is a permanent magnet in the needle tip, yet they are highly steerable by virtue of an external magnetic field. The design and actuation of a magnetic-tip steerable needle for guiding a DBS electrode insertion was presented in [16], and its functionality was demonstrated in an agar brain tissue phantom. The design followed curved trajectories under direct human operator control and executed trajectories with multiple targets.

In their 2006 seminal paper, Webster et al. showed that a bevel-tip needle could be represented with a bicycle model with a fixed steering angle [17]. Slight variations and modifications to the needle model and design have been investigated, including augmenting the model with a steerable angle [18] and, recently, magnetic control of a flexible needle using the bicycle model modified to work with magnetic inputs [19]. However, the bicycle model does not necessarily represent the physics of a magnetic-tip needle in soft tissue. It is subject to a no-slip condition that constrains the velocity of the needle wire and tip in the direction perpendicular to their orientation. This can be an effective constraint on needle tip steering in stiff tissue, but has been found to be violated for magnetic needle insertion in the softer brain tissue phantom [16]. Thus, despite its use in previous applications, the scope of the bicycle model may be limited when applied to magnetic needle steering, particularly in the brain.

Steerable needle models typically include parameters containing the physical properties of the system that dictate needle–tissue interaction. In order to implement a model, these parameters must be identified. Many models directly parameterize tissue and needle properties such as stiffness, elastic modulus, and cutting and friction forces [20,21]. This often requires a specific setup and sensors to explore the tissue–needle interaction. Other models use more abstract parameters that indirectly encode tissue parameters, which are often identified by performing a calibration-specific testing protocol using a theoretically derived best-fit optimization on the calibration data [16,17,22]. These needle model parameterization approaches do not incorporate the identification of measurement or process noise covariances, which precludes optimal model implementation.

Recently, the authors presented a physically motivated magnetic needle model (MNM) specifically for magnetic needle steering in soft tissue and an algorithmic parameter identification methodology for steerable needle models in general [23]. The experimental validity of the model was left undetermined, while the parameter identification methodology was validated in simulation. Building from this prior work, the three primary research contributions of this paper are: (1) algorithmically parameterizing two steerable needle models using the framework from [23] with empirical data, (2) presenting a generalizable method to use the parameterization results to assess which model better fits the empirical data, and (3) demonstrating that the MNM provides a simplified and more physically correct model than the state-of-the-art bicycle needle model (BiNM) in artificial soft tissue.

The remainder of this paper is structured as follows: In Section 2.1, the needle models MNM and BiNM used for parameterization are presented. In Section 2.2, the expectation–

maximization (EM) technique presented in [23] to identify the unknown model parameters, as well as process and measurement noise covariances is summarized and updated. Section 2.3 describes the experimental methods for collecting data ex vivo in agar, while Section 3 provides the results of running the EM algorithm with both the MNM and BiNM to parameterize the models on the experimental data. Section 4 discusses the parameterization results and directly compares the two models' results.

## 2. Methods

### 2.1. Needle Models

The needle design, control, and MNM model were adapted from [23]. The needle contains a nonmagnetic flexible trailing wire, permanent magnet ball joint, and permanent magnet tip, as diagrammed in Figure 1. The needle model states are overlaid in Figure 1, with the heading of the wire indicated by 3D Cartesian unit vector $\mathbf{h_w}$, the position of the ball by $\mathbf{p}$, and the direction of the B-field by $\mathbf{B}$. The direction of the magnetic tip is controlled with an external magnetic field (B-field), and the needle is inserted using a motorized linear advancer at the proximal end of the trailing wire.

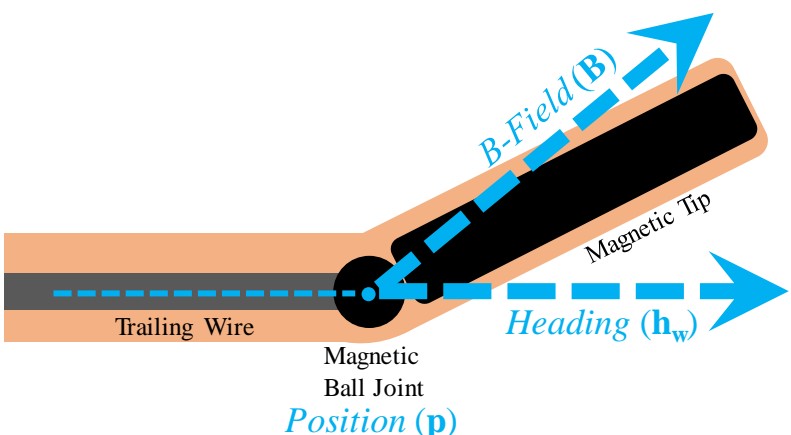

**Figure 1.** Magnetic-tip needle diagram adapted from [23] with states labeled in blue.

A deliberate choice was made to use the 2-degree-of-freedom (DOF) B-direction unit vector as a model state rather than including B magnitude for a 3-DOF B-field vector. As investigated in [16], there exists a minimum turn radius ($100 \pm 15$ mm) before tissue damage is inflicted. This minimum radius corresponds to a maximum field strength when applied to maximum effect (perpendicular to the needle heading). If the field strength is set constant at this maximum, the turn radius can be completely controlled up to this limit by manipulating the B-direction alone. Thus, the assertion is made that this maximum field strength is used and set constant, enabling a reduced DOF model that excludes the B-field magnitude.

#### 2.1.1. Magnetic Needle Model

The state-space kinematic needle steering model MNM from [23] is shown below, where $c$ is an unknown constant related to curvature, $h(\mathbf{x})$ represents the generic measurement model, and $\mathbf{u}$ indicates control inputs, where $v$ is the scalar insertion velocity and $\omega_{\mathbf{B}}$ is the angular rate of change of the B-field.

$$\dot{\mathbf{x}} = \begin{bmatrix} \dot{\mathbf{p}} \\ \dot{\mathbf{h}}_{\mathbf{w}} \\ \dot{\mathbf{B}} \end{bmatrix} = \begin{bmatrix} \mathbf{h_w}v \\ cv(\mathbf{h_w} \times \mathbf{B}) \times \mathbf{h_w} + \mathbf{h_w}(1 - \|\mathbf{h_w}\|^2) \\ \omega_{\mathbf{B}} \times \mathbf{B} + \mathbf{B}(1 - \|\mathbf{B}\|^2) \end{bmatrix} \tag{1}$$

$$\mathbf{y} = h(\mathbf{x}) \tag{2}$$

$$\mathbf{u} = \begin{bmatrix} v & \omega_{\mathbf{B}}^\top \end{bmatrix} \tag{3}$$

Because measurement models are based on hardware and sensors and not physical dynamics, Equation (2) is expressed in the general form. For details on the specific measurement models used for the hardware in this paper, see Section 2.3.9.

### 2.1.2. Bicycle Model

The development of the MNM was motivated by expected shortcomings of the BiNM when used with a magnetic-tip needle, particularly in soft brain tissue. To assess this premise and identify scenarios where MNM may provide superior performance, the methods and analysis presented in this paper were performed on both the MNM and the BiNM to allow for direct comparison of the models.

Hong et al. presented a seven-state three-input BiNM for use in magnetic applications [19]. Equation (4) shows this model converted to match the nine states and four inputs of the derived MNM so the models can be used interchangeably.

$$\dot{\mathbf{x}} = \begin{bmatrix} \dot{\mathbf{p}} \\ \dot{\mathbf{h}}_\mathbf{t} \\ \dot{\mathbf{B}} \end{bmatrix} = \begin{bmatrix} \mathbf{h_t}v \\ l_{inv}v\frac{(\mathbf{h_t}\times\mathbf{B})\times\mathbf{h_t}}{\hat{\mathbf{B}}^\top\mathbf{h_t}} + \mathbf{h_t}(1-\|\mathbf{h_t}\|^2) \\ \boldsymbol{\omega}_\mathbf{B}\times\mathbf{B} + \mathbf{B}(1-\|\mathbf{B}\|^2) \end{bmatrix} \tag{4}$$

Additionally, the BiNM parameter $l$ is moved to the numerator as $l_{inv}$, which allows it to interact identically to $c$ when parameterizing the model.

The mirroring states, inputs, and parameters of the MNM in the BiNM allow for the direct interchangeability of the models. Thus, while the remainder of the paper is written from the perspective of using the MNM, all methods and analysis can be equivalently performed using the BiNM.

Despite these conversion efforts, one major distinction between the BiNM and MNM remains. In the BiNM, the heading state is the needle tip heading $\mathbf{h_t}$, not the trailing wire heading $\mathbf{h_w}$, as in the MNM. To avoid this potential inconsistency and retain model interchangeability, our system intentionally does not measure the heading directly, but allows the models to internally manage the heading by relying on an extended Kalman filter (EKF) to estimate it.

Aside from the heading state, the other primary difference in the models is the $\hat{\mathbf{B}}^\top\mathbf{h_t}$ denominator term in $\dot{\mathbf{h}}_{\mathbf{h_t}}$ in the BiNM (Equation (4)). This term represents the no-slip singularity condition of the BiNM. When $\mathbf{B}$ is perpendicular to $\mathbf{h_t}$, the model is singular and cannot move. This is a significant shortcoming of the BiNM when used with magnetic needle steering and is further discussed in Section 4.2.

### 2.2. Parameterization

To identify the unknown parameters $c$, process noise covariance $Q$, and measurement noise covariances $R_j$ experimentally, the parameterization methods in [23] were used, including augmenting the unknown constant $c$ (or $l_{inv}$) to the state vector and expectation–maximization. The following minor updates to this approach were also implemented.

The generic discrete nonlinear state-space measurement equation in Equation (5) from [23] can be expanded to include multiple independent measurements and corresponding measurement models and $R$s (indicated with subscript $j$), that are time independent from model updates (denoted by replacing subscript $n$ with $m$), as shown in Equation (6).

$$\begin{aligned} y_n &= h(x_n) + v_n \\ v &\sim \mathcal{N}(0,R) \end{aligned} \tag{5}$$

$$\begin{aligned} y_{j,m} &= h_j(x_m) + v_{j,m} \\ v_j &\sim \mathcal{N}(0,R_j) \end{aligned} \tag{6}$$

These allow for the use of multiple generic measurement models, as seen in Section 2.3.9. This update to EM can be seen in Algorithm 1.

---

**Algorithm 1:** EM using the IEKS E-step and multiple measurement models.

---

**Data:** $Y_N$ and $U_N$

**Result:** Converged parameters $Q$, $R_1 \cdots R_J$, and $\hat{X}_N$

1  Guess parameters $Q$ and $R_1 \cdots R_J$ and initial state ($x_0$) and covariance ($P_0$)

2  **do**

3     E-Step: (IEKS)

4     **do**

5        Find the smoothed distribution estimate $p(X_N|Y_N, U_N, Q, R_1 \cdots R_J)$ by using EKS to find $\hat{X}_N$ and $\hat{P}_N$

6        $x_0 \Leftarrow \hat{x}_{0|N}$

7     **while** $\|\Delta x_0\|$ *between iterations* $> tol_{E-Step}$;

8     M-step:

9     Maximize the likelihood function $L(Q, R_1 \cdots R_J)$ independently for $Q$ and each $R_j$ to obtain:
$$Q \Leftarrow \underset{Q}{\mathrm{argmax}}\, L(Q, R_1 \cdots R_J)$$

      **for** *j=1:J* **do**

10        
$$R_j \Leftarrow \underset{R_j}{\mathrm{argmax}}\, L(Q, R_1 \cdots R_J)$$

11     **end**

12  **while** $\|\Delta Q\| + \|\Delta R_1\| + \cdots + \|\Delta R_J\|$ *between iterations* $> tol_{EM}$;

---

In addition, in this EM implementation, the more generic iterated EKS (IEKS) [24,25] replaces the repeated E-step introduced in [23] to converge initial conditions in the nonlinear E-step.

### 2.3. Experimental Setup

#### 2.3.1. Manipulation System

The hardware setup used for these experiments is shown in Figure 2. A custom-built 3-pair nested Helmholtz coil system generates a homogeneous 3D magnetic field inside the workspace. The workspace is $140 \times 140 \times 50$ mm, and the maximum total field strength is 28 mT. A custom needle advancer linearly inserts the needle using tensioned rollers driven by a geared-down DC motor and controller (Maxon Motor, Sachseln, Switzerland), capable of insertion speeds up to 5 mm/s. A support sheath abuts the tissue phantom to prevent the needle from buckling during insertion.

#### 2.3.2. Visual Position Feedback

A 2D xy-plane needle tracking is performed by an overhead BFS-U3-51S5C camera (Point Grey, Richmond, BC Canada) mounted above the tissue phantom, as seen in Figure 2. The camera captures images at 6 Hz, which are processed in real-time using the OpenCV library [26] to identify the needle tip. First, the image is converted to 8-bit grayscale. Then, a manual mouse click on the needle image in the GUI is used to set the adaptive binary threshold to 5 above the pixel value to identify the needle rectangle silhouette from the background. OpenCV open then close operations are performed, and then, the needle silhouette is identified using OpenCV findContours. The centroid of this rectangle is the center position, and the direction of the long axis of this rectangle is the tip heading. Finally, the adaptive threshold value is updated from the current needle center pixel value.

The needle tip's center xy position and xy direction are transformed to the world frame and extracted. The state position **p** is calculated by offsetting half the needle length from the center position in the direction of the tip heading towards the trailing wire. Camera calibration is performed before testing to generate the transform from image frame pixels to world frame meters. The $800 \times 668$ pixel images correspond to a $90 \times 75$ mm field of view of the workspace, resulting in a pixel resolution of about 0.1 mm.

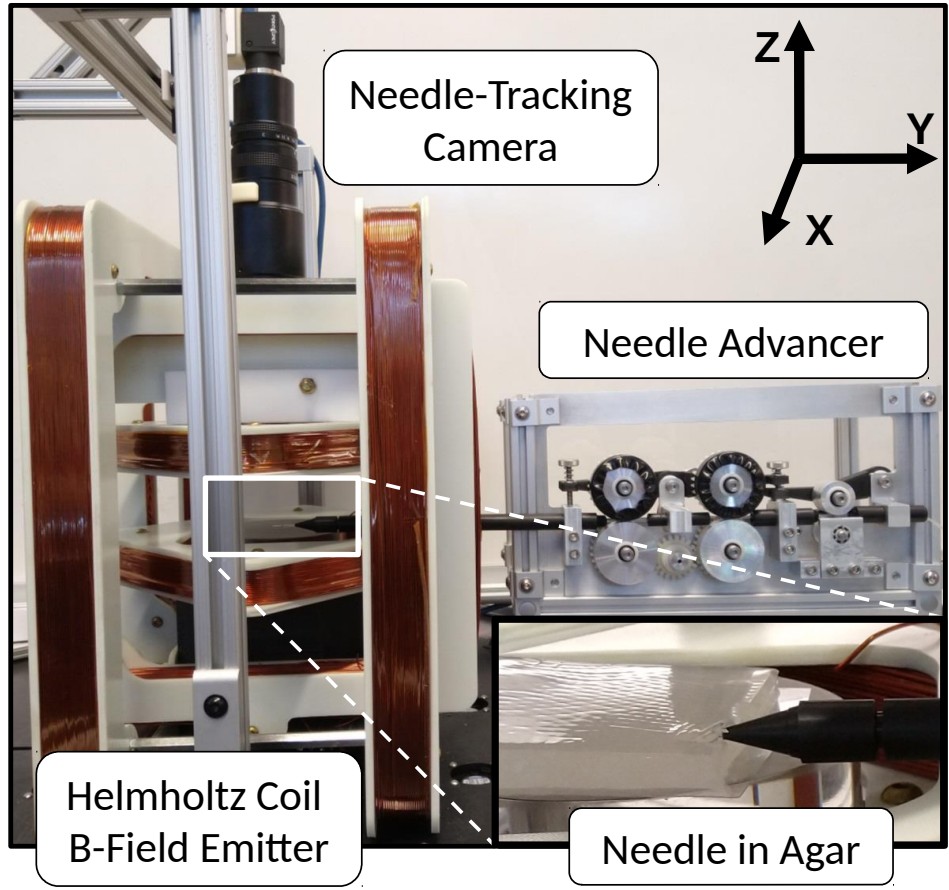

**Figure 2.** Hardware setup: needle advancer inserting needle into workspace via support sheath inside the Helmholtz coil 3D B-field emitter system with overhead camera to track needle. "Needle in Agar" shows magnified view of workspace with agar tissue phantom and abutted needle support sheath. The needle is not visible, but passes through the support sheath into the phantom.

### 2.3.3. Brain Tissue Phantom

Brain tissue is simulated with an agar hydrogel phantom at a 0.6% concentration by weight [27]. For each batch, the dry agar powder (Landor Trading Co, Williamsport, PA, USA) is mixed with distilled water, heated to 100 °C, and stirred until dissolved. After cooling at room temperature for 10 min, the mixture is poured into a mold 14 mm thick and cooled at 4 °C for at least 3 h to solidify and reach thermal equilibrium. Immediately before use, the agar is cut into four 80 × 80 mm pieces to fit in the camera field of view and mounted onto transparent slides.

### 2.3.4. Needle Design and Fabrication

Petruska et al. described magnetic needle design considerations for use in DBS electrode placement surgery [16]. The needle used in this paper is adapted from that design. As shown in Figure 3, the needle wire consists of a 0.25 mm diameter nitinol wire core (McMaster-Carr, USA. Elastic Modulus = 83 GPa) inside a 0.76 mm I.D. 1.65 mm O.D. silicone rubber sheath (McMaster-Carr, USA. Elastic Modulus = 0.05 GPa). The needle tip is a 1 mm diameter by 7 mm length axially magnetized NdFeB cylindrical permanent magnet (HKCM Engineering, Eckernförde, Germany). Behind the needle is a 1 mm-diameter magnetized ball (HKCM Engineering, Germany). The nitinol wire and ball joint are adhered to each other and the silicone sheath using Super Glue (The Gorilla Glue Company, Cincinnati, OH, USA). The tip is capped with a Gorilla Glue hemisphere (The Gorilla Glue Company, USA).

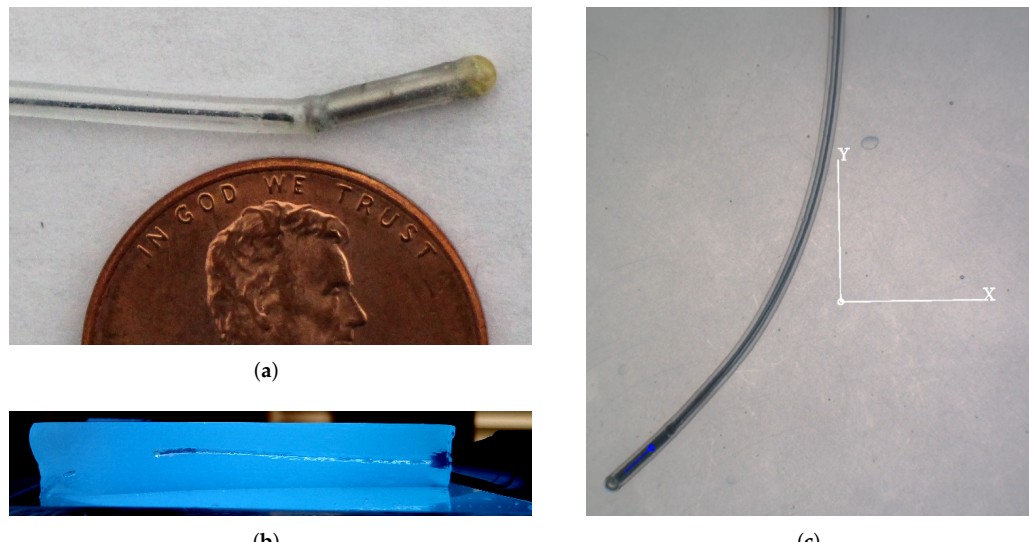

(a)

(b)                 (c)

**Figure 3.** (**a**) Needle trailing nitinol wire, magnetic ball joint, and magnetic tip, all sheathed in a silicone rubber sheath. Representative views of the completed needle insertion trajectory (**b**) cross-sectioned through the yz-plane, and (**c**) in agar from overhead tracking camera.

### 2.3.5. Trajectories

For the purpose of model identification, the simulation results demonstrate that a planar trajectory is sufficient. Thus, while the MNM is 3D-capable, the experiments were performed using a 2D trajectory. This allows for a simplified hardware setup and experimental design. A single overhead camera accurately tracks the position in the xy-plane; the agar thickness was limited, and intuitive 2D test trajectories were used.

Trajectory variation was not expected to significantly affect parameterization, so two disparate experimental trajectories were considered sufficient. Trajectory 1 (T1) (Figure 4) is designed to provide path and magnetic control input variation for realistically expected needle performance and behavior in application. T1 performs a downward S-turn via a constantly increasing $\omega_{\mathbf{B}z}$ that inverts partially through the trajectory. Trajectory 2 (T2) (Figure 4) attempts to achieve the minimum curvature by turning constantly in the same direction. T2 also provides a convenient case where the models diverge, as shown in Section 4.2.2.

Both trajectories are run at a constant insertion velocity of 2.0 mm/s for 27 s to maximize the use of the camera's field of view. The initial position is centered ($x = 0$) at the top ($+y$) of the workspace with the needle inserted in the down ($-y$) direction. Both trajectories are mirrored to run in both $\pm x$ directions to analyze trajectory and direction results independently.

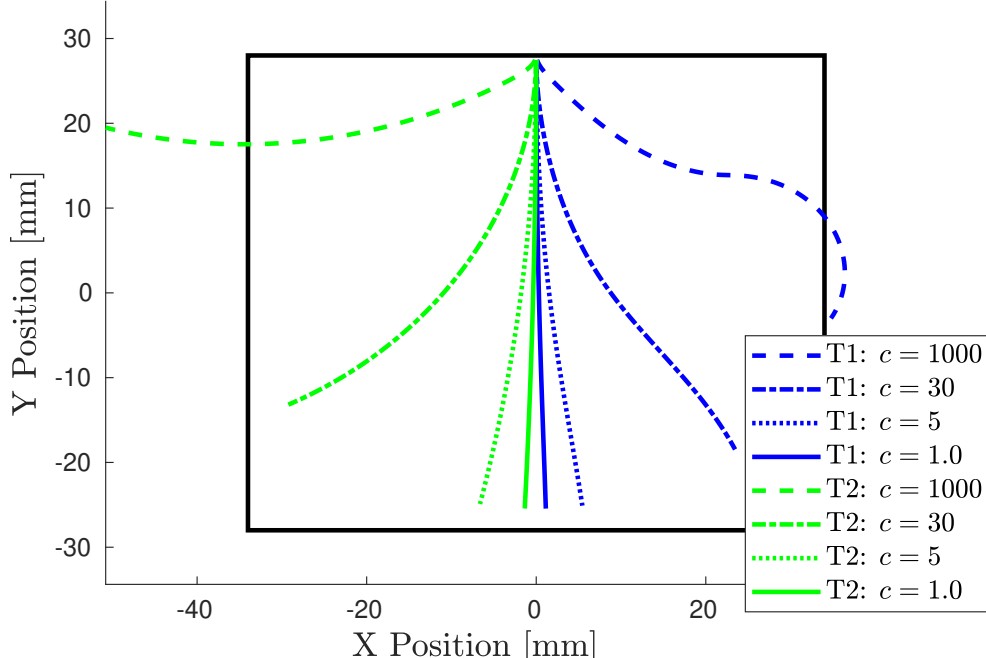

**Figure 4.** Predicted experimental trajectories using MNM with various expected $c$ values. ($c = 1000$ is not expected, but is depicted to show the limit case.) T1 pictured in positive direction and T2 pictured in negative direction for visual clarity. The black square represents the camera field of view.

### 2.3.6. Planar Control

In a 2D analysis, the trajectory is assumed perfectly planar, so deviation out-of-plane reduces the accuracy of the results. The experimental 2D trajectories were implemented using $\omega_{\mathbf{B}z}$ control only ($\omega_{\mathbf{B}x} = \omega_{\mathbf{B}y} = \mathbf{B_z} = 0$), and without active magnetic control, the needle can drift freely in the z-plane. To maintain a planar trajectory when collecting data, we implemented proportional control on $\mathbf{B_z}$, as shown in Equation (7). This control was based on the relative opacity of the needle in the agar from the initial insertion point, as measured by the 0–255 grayscale pixel values at the tip of the needle from the overhead camera. The gain of 2.0 mT per pixel value was set empirically to provide the least deviation of the needle in the z-plane, and the $\mathbf{B_z}$ control input was saturated at the $\pm 28$ mT hardware limit. Figure 3 shows a representative cross-section of a complete insertion trajectory. Cross-sections were performed on the last trial in each agar piece, and all vertical displacement from planar was found to fall within 3 mm.

$$\mathbf{B_z} = 2.0(\text{opacity}_{current} - \text{opacity}_{initial}) \tag{7}$$

### 2.3.7. Test Procedure

Three batches of agar were made, and each batch was cut into four pieces. Four trials were performed in each piece, where, in each trial, the needle was inserted into a different edge of the piece, resulting in a total of 48 individual trials.

### 2.3.8. Field Strength

As discussed in the model section, the application intent is to select a constant field strength that optimally balances the max curvature without damaging the tissue [16]. However, this field strength was not known a priori, so three field strengths were tested: 10, 19 and 28 mT. This also allows for the analysis of the field strength to $c$ relationship, as they should be positively correlated.

2.3.9. Measurement Models

In this system, two measurement models are used: one for the camera using **p** and one for the Helmholtz coils using **B**. The overhead camera measures the position **p** of the magnetic tip directly, so the measurement model is the identity:

$$\mathbf{y}_1 = h_{\mathbf{p}}(\mathbf{x}) = \mathbf{p} \tag{8}$$

The camera can measure $\mathbf{h_t}$ because it can identify the direction from the straight needle tip, but it does not measure $\mathbf{h_w}$ because it is not able to accurately identify a line tangential to the distal end of the trailing wire. Thus, for consistency in the analysis between models, in addition to the reasons listed in Section 2.1.2, a camera $\mathbf{h_t}$ measurement is not used, and instead, both models' headings are estimated in the EKF.

The positional measurement model was enhanced to include a minimum noise threshold to help prevent overfitting. This minimum was set at the sensor limit based on the 0.1 mm pixel resolution of the camera. Dividing the resolution by two for bidirectionality and converting units resulted in a minimum positional measurement noise covariance threshold of $2.5 \times 10^{-9}$ m$^2$.

The Helmholtz coil approximates a spinning magnetic field as described by the input $\omega_{\mathbf{B}}$ by updating static field targets at 300 Hz. Discrete measurements were taken directly from this input at 18 Hz. Thus, the B-field measurement model is the identity:

$$\mathbf{y}_2 = h_{\mathbf{B}}(\mathbf{x}) = \mathbf{B} \tag{9}$$

The two measurement models in Equations (8) and (9) correspond to measurement noise covariances $R_{\mathbf{p}}$ and $R_{\mathbf{B}}$, which are both maximized in the M-step of Equation (1).

We recognize that the B-field measurements are synthetic, since they come directly from the input. The B-field is difficult to measure independently in this system because the workspace is too spatially constrained to fit a magnetometer sensor, as it would interfere with at least one of the following: camera view, needle advancer, or tissue phantom. Fortunately, it is not necessary to measure the B-field, because the Helmholtz coils were calibrated by a magnetometer before each series of trials. This resulted in an $R_{\mathbf{B}}$ that did not capture random measurement noise; it only captured quantization error resulting from differences in the static field update and measurement frequencies, which should be negligible. These synthetic measurements were used because they improve the algorithm's accuracy, and $R_{\mathbf{B}}$ was still identified and reported for completeness. However, the analysis of $R_{\mathbf{B}}$ was omitted, because it did not provide meaningful information about algorithmic performance or model differences.

## 3. Results

Four trials experienced experimental error, and the data were discarded, resulting in a total of 44 successful individual trials. EM as described in Algorithm 1 was run on each trial of experimental data for both the MNM and BiNM with a limit of 100 iterations. An example trajectory result is shown in Figure 5. EM converged on all trials with the MNM, but diverged on three trials when using the BiNM due to breaching the singularity. Those three trials were excluded from results of both models for consistency.

The EM algorithm was parameterized $c$ with an average of 5.3, 9.5, and 11.9 m$^{-1}$ for the three different field strengths tested and standard deviations (SDs) of 3.2, 4.8, and 5.6 m$^{-1}$, respectively, as shown in the first column of Table 1. Similarly, $l_{inv}$ was parameterized with an average of 1.6, 3.5, and 4.9 m$^{-1}$ for the three different field strengths tested and standard deviations of 1.6, 3.0, and 3.4 m$^{-1}$, respectively, as shown in the first column of Table 2.

Since $Q$ and $R$ are matrices, we reduced them to single average value representations by dividing their trace by the measurement dimension to facilitate our analysis and comparison. As the trace is the sum of the eigenvalues, it provides a measure of the total variation, and dividing provides an average variation per 3D dimension. $R_{\mathbf{p}}$ and $R_{\mathbf{B}}$ are each reduced using their trace to a single representative value because each measures only a single unit:

position variance (m$^2$) and B-field direction variance (unit vector $^2$), respectively. Note from the squared units that the $Q$ and $R$ values are reported as variances in all tables and figures, so to interpret variation with respect to states, the square root must be taken.

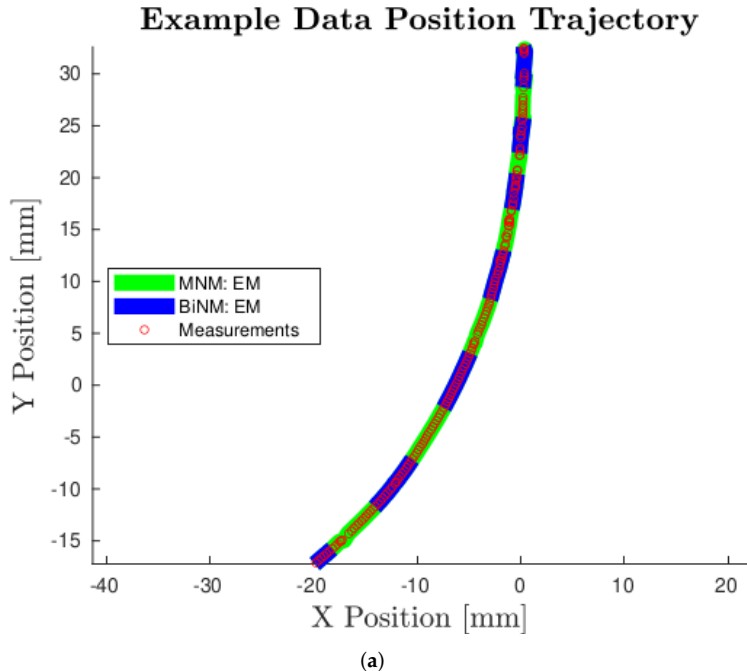

(**a**)

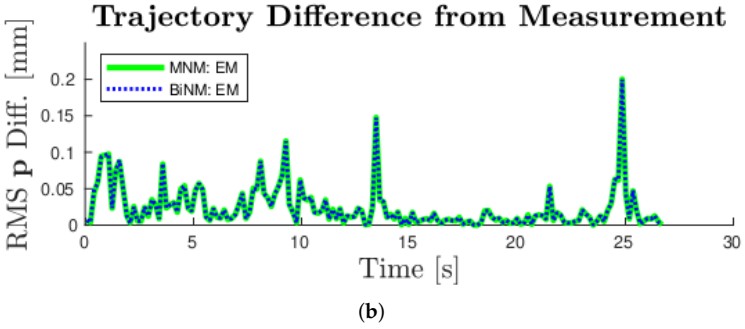

(**b**)

**Figure 5.** (**a**): Example experimental results for Trajectory 2 in the -X-direction with a field strength of 28 mT. Converged EM position state estimates are shown for both models. (**a**) shows the trajectories' x-y position and experimental measurements, while (**b**) shows RMS position difference of the EM trajectories from the measurements. As seen in both (**a**) and (**b**), these trajectories are nearly indistinguishable, demonstrating strong agreement between models for these data.

**Table 1.** MNM parameterization results.

|  | $c$ | | | $tr(Q_{\mathbf{p}})$ | $tr(Q_{\mathbf{h}})$ | $tr(Q_{\mathbf{B}})$ | $tr(R_{\mathbf{p}})$ | $tr(R_{\mathbf{B}})$ |
|---|---|---|---|---|---|---|---|---|
| @Field Strength | 10 mT | 19 mT | 28 mT | | | All | | |
| Mean | 5.3 | 9.5 | 11.9 | $1.10 \times 10^{-7}$ | $6.65 \times 10^{-4}$ | $7.32 \times 10^{-9}$ | $2.70 \times 10^{-9}$ | $5.6 \times 10^{-12}$ |
| SD | 3.2 | 4.8 | 5.6 | $1.23 \times 10^{-7}$ | $6.58 \times 10^{-5}$ | $1.18 \times 10^{-8}$ | $6.58 \times 10^{-10}$ | $5.8 \times 10^{-12}$ |
| SD as Traj. Err. (mm) | 0.002 | | | 0.022 | 0.003 | 0.052 | $2 \times 10^{-13}$ | NA [1] |

[1] Analysis of $R_{\mathbf{B}}$ is omitted as explained in Section 2.3.9.

**Table 2.** BiNM parameterization results.

| | $l_{inv}$ | | | $tr(Q_\mathbf{p})$ | $tr(Q_\mathbf{h})$ | $tr(Q_\mathbf{B})$ | $tr(R_\mathbf{p})$ | $tr(R_\mathbf{B})$ |
|---|---|---|---|---|---|---|---|---|
| @Field Strength | 10 mT | 19 mT | 28 mT | | | All | | |
| Mean | 1.6 | 3.5 | 4.9 | $1.10 \times 10^{-7}$ | $6.40 \times 10^{-3}$ | $7.37 \times 10^{-9}$ | $2.72 \times 10^{-9}$ | $4.4 \times 10^{-12}$ |
| SD | 1.6 | 3.0 | 3.4 | $1.23 \times 10^{-7}$ | $2.04 \times 10^{-2}$ | $1.18 \times 10^{-8}$ | $7.68 \times 10^{-10}$ | $3.7 \times 10^{-12}$ |
| SD as Traj. Err. (mm) | | 0.114 | | 0.302 | 0.360 | 0.200 | 0.001 | NA [1] |

[1] Analysis of $R_\mathbf{B}$ is omitted as explained in Section 2.3.9.

$Q$ contains three $3 \times 3$ sub-matrices corresponding to the three 3D states: position, heading, and B-field direction. Results represent each sub-matrix as a singular value using its trace: $tr(Q_\mathbf{p})$, $tr(Q_\mathbf{h})$, and $tr(Q_\mathbf{B})$, respectively. Complete $Q$ and $R$ matrices' cross-terms were found to be at least one order of magnitude lower than the diagonals, while the diagonals within each $Q$ block and $R$ were found to be within an order of magnitude, so the information lost by this reduced representation should not significantly impact the interpretation. These average parameterized values along with their standard deviations are reported in Tables 1 and 2 for the MNM and BiNM, respectively. These results show that, on the order of a $10^{-12}$ variance of a unit vector, $R_\mathbf{B}$ is minuscule, as predicted in Section 2.3.9, supporting its omission from further analysis.

The spread of the $tr(Q_\mathbf{p})$, $tr(Q_\mathbf{h})$, $tr(Q_\mathbf{B})$, and $tr(R_\mathbf{p})$ data are shown with boxplots in Figure 6. The tight one-sided distribution of $tr(R_\mathbf{p})$ was caused by the minimum noise threshold set at $2.5 \cdot 10^{-9} \, \mathrm{m}^2$. Parameterization of $tr(Q_\mathbf{p})$, $tr(Q_\mathbf{B})$, and $tr(R_\mathbf{p})$ was nearly identical for MNM and BiNM. In contrast, parameterization of $tr(Q_\mathbf{h})$ did not match between models and on average was an order of magnitude higher in the BiNM than in the MNM. Additionally, three extreme high-side outliers occurred in BiNM at $tr(Q_\mathbf{h}) = 0.12$, 0.036, and 0.026 (not pictured in Figure 6 for clarity).

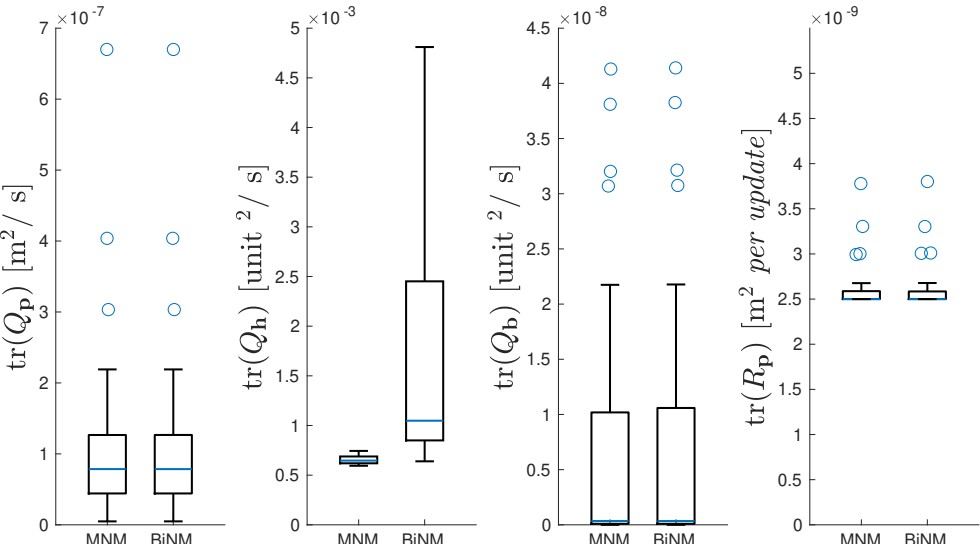

**Figure 6.** Boxplots of all 41 trials' $tr(Q_\mathbf{p})$, $tr(Q_\mathbf{h})$, $tr(Q_\mathbf{B})$, and $tr(R_\mathbf{p})$ for MNM and BiNM. Boxes and whiskers indicate quartiles; blue line indicates median; blue circles indicate outliers. $tr(Q_\mathbf{h})$ shown without outliers for visual clarity.

Ultimately, parameterization was used to improve the predictive accuracy of the models. Thus, positional trajectory sensitivity to the parameters was explored. The last row of Tables 1 and 2 presents a difference in the RMS distance trajectory error that could be expected at one standard deviation from the mean for each parameter. To calculate these values, both T1 and T2 were first simulated nominally using the average parameters from Tables 1 and 2. Then, each parameter was separately inflated and deflated by the ratio of its mean to the standard deviation. The average trajectory error from this nominal trajectory was calculated, and this process was repeated 100 times and averaged. This resulting measure of error provides a physical interpretation of the parameter variation.

## 4. Discussion

The experiments in this paper serve two goals: to use the EM algorithm to identify needle model parameters and to examine differences between the MNM and the BiNM to determine if and when the MNM provides superior performance to the BiNM. Having better model state estimation will ultimately result in improved control of the needle and, thus, more accurate electrode placement in surgery.

### 4.1. Parameter Identification

The EM algorithm converged to a solution for every single trial using the MNM. With the BiNM, a solution was found for all but three trials, and these failed due to the known model issue of approaching the singularity. Additionally, parameterization spread results expressed by the standard deviations in Tables 1 and 2 demonstrate a level of consistency between the solutions.

These variation measures of the $c$, $Q$, and $R$ parameters help to understand the precision of the EM algorithm applied to these needle steering models. However, to practically evaluate the parameterization, we considered the sensitivity of the trajectory error to this variation. A higher trajectory error might indicate that the variance in parameterization was too large, and the model did not capture all significant physical effects, while a lower trajectory error indicates that average parameterization results can be generally applied. As seen in Tables 1 and 2, the trajectory error produced by both models' parameter spread at one standard deviation was submillimeter across all parameters, with the largest being 0.360 mm from $Q_\mathbf{B}$ of the BiNM. This is very promising compared to the 2 mm surgical accuracy target for electrodes in Deep Brain Stimulation [4] and supports using the resulting parameterized model for general needle steering application.

The mean $tr(R_\mathbf{p})$ for MNM as reported in Table 1 was $2.70 \times 10^{-9}$ m$^2$, so the error in the position measurement (expressed as a standard deviation) was 0.05 mm. This is comparable to the camera pixel resolution due to the $tr(R_\mathbf{p})$ distribution falling very near the minimum limit established at the camera resolution. Because both models use the same measurements, the measurement noise should be the same. This is supported in the results, as $tr(R_\mathbf{p})$ closely matches between the models.

In these results, $Q$ is expressed as a continuous variance per second, while $R$ is a per update value that applies to each measurement. Therefore, on average, to prevent $Q_\mathbf{p}$ noise from accumulating larger than $R_\mathbf{p}$, position measurements would need to occur at a frequency of the ratio of the two: 5.2 Hz. Experimental measurements occurred at 6 Hz, which is not coincidentally very similar. This is because the EM algorithm matches model positional error to measurement error by pushing uncertainty into the heading. Since the heading is estimated, it has no measurement to reduce or correct its uncertainty like the position and B-field direction. $Q_\mathbf{p}$ from both models matched, which is reasonable, because both models use position in the same way.

Similarly, $Q_\mathbf{B}$ also matched between the models. $Q_\mathbf{B}$ was extremely small, at $7 \times 10^{-9}$, which correlates with the 0.008 %/s error of a unit vector, or equivalently, a 0.005 °/s error in direction. $Q_\mathbf{B}$ is not expected to experience significant error because no dynamics affect the B-field direction, so it is free to follow the $\omega_\mathbf{B}$ control input precisely.

$Q_{\mathbf{h}}$ is the pivotal element of process noise for two reasons. First, because the heading is estimated, the uncertainty of the models collects in $Q_{\mathbf{h}}$. This makes it a strong indicator of model accuracy. Second, the primary difference between models is how the heading state interacts, which will make it an important aspect of the model comparison in Section 4.2. The average $Q_{\mathbf{h}}$ for the MNM is $6.65 \times 10^{-4}$, which correlates with the 2.6 %/s error of a unit vector, or equivalently, a $1.5°$/s error. The average $Q_{\mathbf{h}}$ for the BiNM was $6.40 \times 10^{-3}$, which correlates with the 8.0 %/s error of a unit vector, or a $4.6°$/s error, which is triple that of the MNM.

### 4.2. Model Comparison

The MNM is presented as an alternative to the current de facto needle steering model, the BiNM. Here, we compare the results from the two models, focusing on areas where they differ, and present why the MNM is the preferred model to use for magnetic needle steering.

#### 4.2.1. Comparing Process Noise

Process noise $Q$ is a measure of error accumulation per time of the difference between the true physical system and the model. When comparing $Q$ between models, the model with a lower $Q$ can be interpreted as more representative of the truth, and thus the better model.

The percent difference of BiNM to MNM noise traces is shown in the first row of Table 3. To ensure no significant differences exist in any term, as was done in calculating the simulation results, full $Q$ and $R$ matrix blocks were compared between models by finding a percent difference between median data matrices using the matrix norm. Results are shown in the second row of Table 3.

Both comparisons resulted in the same trends. $Q_{\mathbf{p}}$, $Q_{\mathbf{B}}$, and $Q_{\mathbf{p}}$ showed negligible differences, while $Q_{\mathbf{h}}$ was much larger in the BiNM than the MNM. This 863% larger $Q_{\mathbf{h}}$ in the BiNM demonstrates a significant disparity between the models and supports the MNM being a more physically representative model of the data than the BiNM.

**Table 3.** The % difference in $Q$ and $R$ of BiNM from MNM.

|  | $Q$ | | | $Rp$ |
|---|---|---|---|---|
|  | **p** | **h** | **B** | Overall |
| Trace | 0.0% | 863% | 0.6% | 0.7% |
| Blockwise | 5.9% | 369% | 2.6% | 0.0% |

To understand the practical effect of this difference in $Q$, Figure 7 presents data boxplots of the trajectory effect as both the average position RMS difference and the final position RMS difference between the two models. All measures of the difference in trajectory as a result of the difference in $Q$ between the models were negligible, with the largest being 0.04 mm. It is possible that with fewer or less accurate measurements where the model would play a greater role, there could be a larger distinction, but in this experimental data, there was no practical difference.

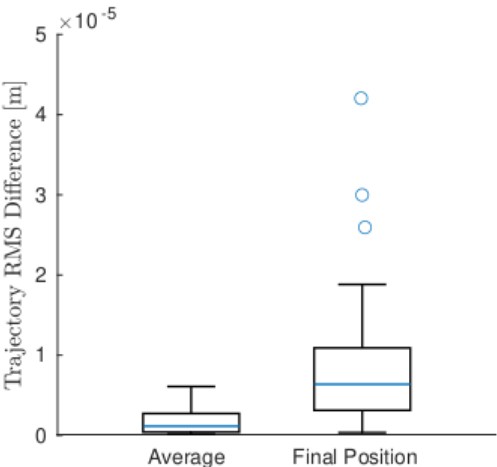

**Figure 7.** Boxplots of all 41 trials' average and final position RMS difference between MNM and BiNM. Boxes and whiskers indicate quartiles; blue line indicates median; blue circles indicate outliers.

### 4.2.2. Bicycle Model Singularity Problem

To test the difference between models, we designed an experimental trajectory scenario where they diverged significantly. The model that more closely matches the data for this scenario will be the more physically realistic model. This trajectory scenario, T2$_{prelim}$, occurred as the BiNM approaches its singularity and is shown in Figure 8.

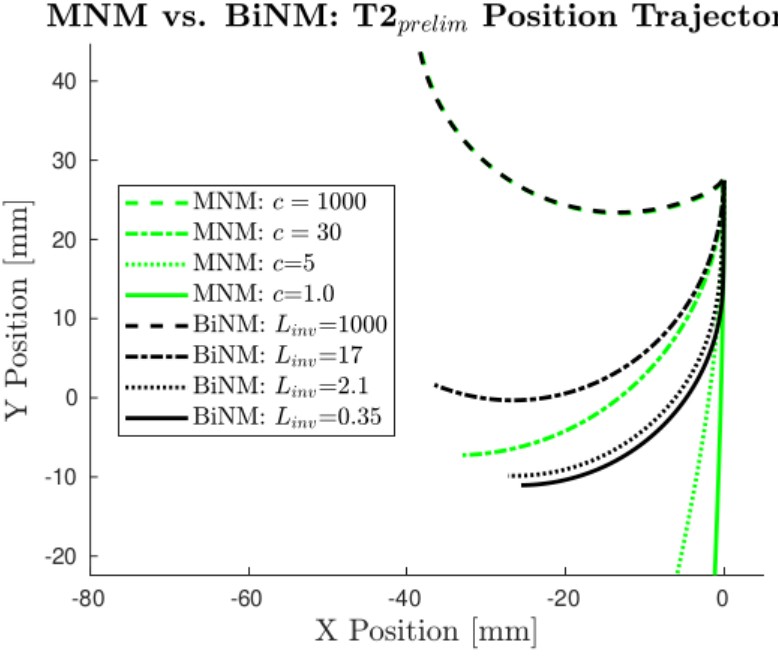

**Figure 8.** Models' predicted position trajectories diverge as they approach the BiNM singularity ($\mathbf{h} \perp \mathbf{B}$) in T2$_{prelim}$. MNM position trajectories are plotted using several realistic $c$ values, with $c = 1000$ also shown as the limit case. Corresponding BiNM trajectories are plotted alongside the MNM's using $L_{inv}$ values manually tuned to match each MNM trajectory as closely as possible.

If the data followed the BiNM, the curvature would increase when approaching the singularity, effectively causing a "pulling" effect of the BiNM that keeps the model out of the singularity. On the other hand, the MNM allows crossing $\mathbf{h} \perp \mathbf{B}$ without issue, and so predicts a straighter trajectory. This effect becomes more pronounced at lower $c$ and $L_{inv}$ values, as $c$ allows an even less capable turning rate, thus reaching $\mathbf{h} \perp \mathbf{B}$ earlier, while the BiNM forces the heading to stay out of the singularity, regardless of $L_{inv}$.

Preliminary data were collected using T2$_{prelim}$ and found to most closely match the MNM. As shown in Figure 9, the measurements better followed the predicted open-loop MNM trajectory, thus violating the singularity in the BiNM, and diverged from the predicted open-loop BiNM trajectory.

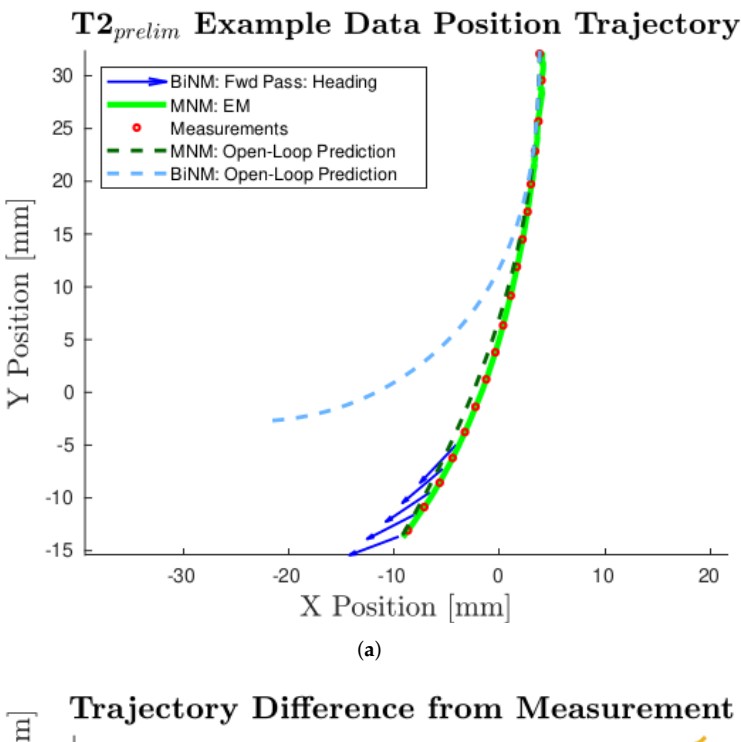

(a)

(b)

**Figure 9.** Example experimental results for T2$_{prelim}$ in the -X-direction with a field strength of 28 mT. Converged EM position state estimate is shown for the MNM. Predicted open-loop position trajectories are shown for both models, using average data $c$ and $L_{inv}$ values at 28mT from Tables 1 and 2, respectively. (**a**) shows the trajectories' x-y-position and experimental measurements (every 8th interval for visual clarity), while (**b**) shows the RMS position difference of each calculated trajectory from the measurements. As seen in both (**a**) and (**b**), while the MNM open-loop prediction agrees with the data, the BiNM prediction diverges significantly. The heading state estimate of the EKF forward pass of the BiNM is also plotted in (**a**) as it became increasingly incongruous with the positional trajectory data.

An EKF forward pass of the BiNM is overlaid, which shows its heading diverging from the trajectory in a nonsensical way so that the model does not enter the singularity. This can be seen more starkly by plotting $\mathbf{h}^\top \mathbf{B}$ directly (Figure 10). The EM MNM estimate is shown crossing the singularity ($\mathbf{h}^\top \mathbf{B} = 0$) in a continuous smooth line, while the forward pass BiNM estimate follows a similar trajectory until the singularity. There, it has a sharp slope discontinuity and remains just outside the singularity, predicting exceedingly unreasonable headings, as was seen in Figure 9a, and diverging from the measurements, as seen in Figure 9b. When attempting to refine the BiNM estimate using multiple EM passes, the

algorithm became unstable because it cannot reconcile the position measurements with the BiNM while avoiding the singularity.

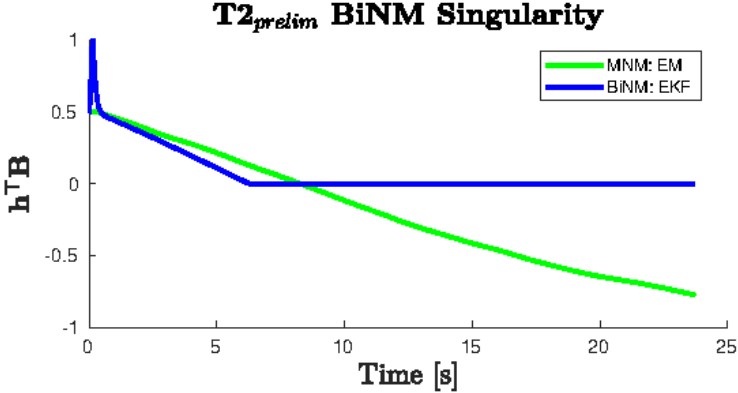

**Figure 10.** $\mathbf{h}^\top \mathbf{B}$ of the T2$_{prelim}$ example data from Figure 9 shows a discontinuity in the BiNM at the singularity ($\mathbf{h}^\top \mathbf{B} = 0$). The EKF forward pass of the BiNM does not cross the singularity boundary. It instead persists just above for the remainder of the trajectory, while the MNM continuously passes through the singularity without disruption.

Because these data were unusable for the parameterization of the BiNM using the EM algorithm, T2$_{prelim}$ had to be modified to avoid the BiNM singularity. This resulted in the experimental T2, which is shown in Figure 11. However, avoiding the BiNM singularity necessarily resulted in model trajectories that did not diverge, because divergence occurred near and as a result of the singularity. Thus, in comparing parameterization results for the purpose of identifying the more physically accurate model, as in Section 4.2.1, the models predicted similar results, and only small differences were found, which did not significantly affect the resulting estimated trajectories. However, it must be recognized that while in this limited region, the models were similar, it is more important that, where they would not be similar, the BiNM cannot be used at all, because it failed due to its singularity.

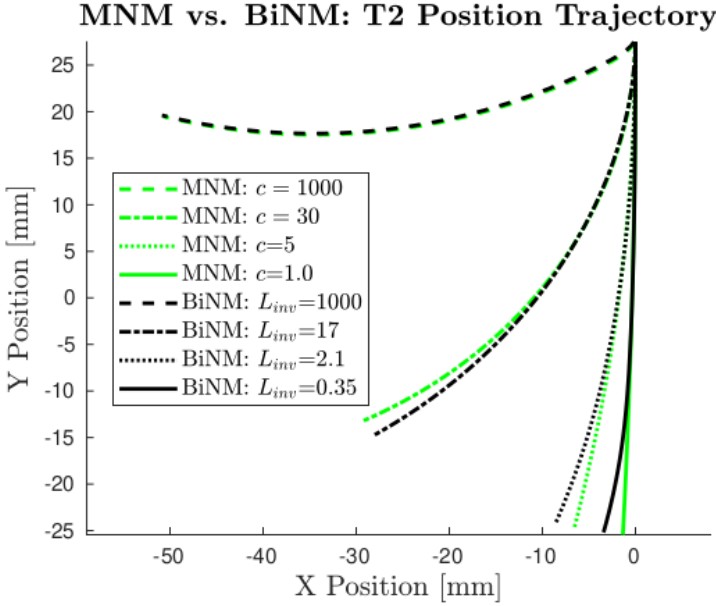

**Figure 11.** Models do not appreciably diverge in the predicted T2 trajectory (T2$_{prelim}$ modified to avoid the singularity). MNM position trajectories are plotted using several realistic $c$ values, with $c = 1000$ also shown as the limit case. Corresponding BiNM trajectories are plotted alongside the MNM's using $L_{inv}$ values manually tuned to match each MNM trajectory as closely as possible.

### 5. Conclusions

We derived a physically motivated needle steering model and demonstrated the use of an EM algorithm with IEKS to identify the unknown model curvature parameters and process and measurement noise covariances, both in simulation and experimentally. The simulation analyzed the convergence envelope and accuracy of the model parameters based on the variables themselves, as well as their initial guesses. The parameters consistently converged, and under expected conditions, parameterization reduced the RMS position trajectory error to 0.81 mm, which compares favorably to the approximately 2 mm accuracy that is required in the final placement of electrodes in DBS [4]. These simulation results support the application of this EM algorithm in identifying $c$, $Q$, and $R$ experimentally.

We collected experimental data in an agar brain tissue phantom using open-loop linear needle insertion and B-field steering inputs. These data were used to parameterize both our derived magnetic needle model and the commonly used bicycle needle model. Parameterization converged for all trials using the MNM, but the BiNM failed to converge with data near its singularity. Variation in $c$ and $l_{inv}$ curvature parameterization, when averaged for each field strength tested, did not express a significant effect on positional trajectory error. Variation in $Q$ and $R$ parameterization also did not manifest a significant effect on positional trajectory error, suggesting the experimental results were sufficiently accurate to use the parameterized models in future applications, such as prediction and control. The next step towards clinical application is developing a controller for this system. The physically determined $c$, $Q$, and $R$ parameters will be needed to apply and demonstrate needle steering control in a physical system.

The MNM parameterized a lower $Q$ than the BiNM in the experimental results, indicating that it more accurately models needle steering in soft brain tissue. For data not near the singularity however, there was no practical difference between the models, as the difference in the position trajectories was negligible. Therefore, if care is taken to avoid the singularity, either model can be used with similar success. However, being near the singularity is not only a likely scenario: it is desirable due to its strong influence on steering. In this case, the BiNM model breaks down and cannot be algorithmically parameterized, because it fails to represent reality. Given that no such limitations exist in the MNM and that it better models needle steering in all cases tested, our results support the use of the MNM over the BiNM in all magnetic needle steering applications.

**Author Contributions:** Conceptualization, R.L.P. and A.J.P.; data curation, R.L.P.; formal analysis, R.P.; funding acquisition, A.J.P.; Investigation, R.P.; methodology, R.L.P.; project administration, A.P.; software, R.L.P.; supervision, A.J.P.; validation, R.L.P.; visualization, R.L.P.; writing—original draft, R.L.P.; writing—review and editing, A.J.P. All authors have read and agreed to the published version of the manuscript.

**Funding:** This research was funded by the Boettcher Foundation's Webb-Waring Biomedical Research Program, Grant Number CSM PROP 17-0522.

**Institutional Review Board Statement:** Not applicable.

**Informed Consent Statement:** Not applicable.

**Data Availability Statement:** The data presented in this study are openly available in FigShare at https://doi.org/10.6084/m9.figshare.16733233 (accessed on 4 February 2022).

**Conflicts of Interest:** The authors declare no conflict of interest.

### Abbreviations

The following abbreviations are used in this manuscript:

DBS     Deep Brain Stimulation
MNM    magnetic needle model
BiNM    bicycle needle model

EKS     extended Kalman smoother
IEKS    iterated extended Kalman smoother

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
