# Peer review of "Empirically Comparing Magnetic Needle Steering Models Using Expectation-Maximization"

_robotics, doi:10.3390/robotics11020049_

Round 1

Reviewer 1 Report

The authors of the article report an experimental identification of model parameters for two different steerable needle models i.e. bi-cycle needle model, and magnetic needle model employing an expectation-maximization (EM) algorithm. They also examine two models, concluding that the magnetic needle model is better compared to the other one for modeling soft brain tissue needle entry.

Overall, this manuscript reads well and provides insights into needle steering control for surgical interventions in the brain. As such, this reviewer recommends that this manuscript could be published in Robotics if the following comments could be reasonably addressed:

  • However, the bicycle model does not necessarily represent the physics of a magnetic-tip needle in soft tissue. It is subject to a no-slip condition that constrains the velocity of the needle wire and tip in the direction perpendicular to their orientation. This can be an effective constraint on needle tip steering in stiff tissue but has been found to be violated for magnetic needle insertion in the softer brain-tissue phantom [16]. Thus, the scope of the bicycle model may be limited when applied to magnetic needle steering, particularly in the brain”. Since it’s plausible that the use of the bi-cycle needle model is limited in the case of brain tissue, it would be better to explain in the paper why yet they choose the bi-cycle model to compare with the magnetic needle model in this specific case.
  • The authors are suggested to mention elaborately the critical part such as timing, temperature, physical dimensions, etc for preparing brain tissue phantom, and needle design and fabrication. For example, what will happen if someone tries to cool down at 0°C, is it critical to cool the sample down specifically at 4°C?
  • “T1 performs a downward S-turn via a constantly increasing wBz that inverts partially through the trajectory. Trajectory 2 (T2) (Fig. 4) attempts to achieve the minimum curvature by turning constantly in the same direction.” Could the author please provide an explanation of why T1 and T2 differ from each other in the first place?
  • Does the trajectory depend upon the insertion velocity? If so, could the author please demonstrate their dependency?
  • The reviewer suggests a careful reading to improve the English of the paper including some typos, for example ‘Means’ instead of ‘Meas’ in Figure 5(a).

Reviewer 2 Report

  The paper discusses the identification of model parameters for steerable needle models, using the expectation-maximalization algorithm. This work also aimed to examine differences between two needle models: the MNM and BiNM. The paper is written in good good English, and it is adequately structured.  The authors provide a throughout literature review and a well-structured introduction on the aim of the paper. Methods are adequately described and the models are presented within a satisfying depth.  Images are of good quality, well-captioned and referenced to.  Results are discussed in a structured manner, the Discussion is detailed and structured with the outlook of further improvement methods.  Overall, the article is well-written, results are relevant and structured.    Question to the authors: in the past years, parameter identification by AI-based methods, especially neural networks and genetic algorithms has gained much attention. How would using these methods affect the methodology of parameter estimation in this particular project? How would this method fair agains free floating/swimming microrobots in the magnetic field? E.g., /doi.org/10.1021/nl901869j  In terms of clinical applicability, TRL level and Level of Autonomy, where do the authors put thier system on the scale presented here: DOI: 10.1109/TMRB.2019.2913282  
